# A Comparative Analysis of Usual- and Gastric-Type Cervical Adenocarcinoma in a Japanese Population Reveals Distinct Clinicopathological and Molecular Features with Prognostic and Therapeutic Insights

**DOI:** 10.3390/ijms26157469

**Published:** 2025-08-01

**Authors:** Umme Farzana Zahan, Hasibul Islam Sohel, Kentaro Nakayama, Masako Ishikawa, Mamiko Nagase, Sultana Razia, Kosuke Kanno, Hitomi Yamashita, Shahataj Begum Sonia, Satoru Kyo

**Affiliations:** 1Department of Obstetrics and Gynecology, Faculty of Medicine, Shimane University, Izumo 693-8501, Japan; farzanashormi99@gmail.com (U.F.Z.); hasibulsohel1167@gmail.com (H.I.S.); m-ishi@med.shimane-u.ac.jp (M.I.); kanno39@med.shimane-u.ac.jp (K.K.); memedasudasu1103@gmail.com (H.Y.); sbsonia1995@gmail.com (S.B.S.); 2Department of Obstetrics and Gynecology, East Medical Center, Nagoya City University, Nagoya 464-8547, Japan; 3Department of Organ Pathology, Faculty of Medicine, Shimane University, Izumo 693-8501, Japan; mami55@med.shimane-u.ac.jp; 4Department of Legal Medicine, Faculty of Medicine, Shimane University, 89-1 Enya-Cho, Izumo 693-8501, Japan; sultana@med.shimane-u.ac.jp

**Keywords:** gastric-type cervical adenocarcinoma, usual-type cervical adenocarcinoma, prognostic biomarker, ARID1B, immune checkpoint inhibitor (ICI) therapy

## Abstract

Gastric-type cervical adenocarcinoma (GCA) is a rare and aggressive subtype of cervical adenocarcinoma. Despite its clinical significance, its molecular carcinogenesis and therapeutic targets remain poorly understood. This study aimed to compare the clinicopathological, immunohistochemical, and molecular profiles of GCA and usual-type cervical adenocarcinoma (UCA), exploring prognostic and therapeutic biomarkers in a Japanese population. A total of 110 cervical adenocarcinoma cases, including 16 GCA and 94 UCA cases, were retrospectively analyzed for clinicopathological features, and a panel of immunohistochemical markers was assessed. Sanger sequences were performed for the *KRAS*, *PIK3CA*, and *BRAF* genes, and survival and clinicopathological correlations were assessed using Kaplan–Meier and Cox regression analyses. GCA was significantly associated with more aggressive features than UCA, including lymph node involvement, advanced FIGO stages, increasing recurrence rate, and poor survival status. High ARID1B expression was observed in a subset of GCA cases and correlated with worse progression-free and overall survival. Additionally, PD-L1 expression was more frequent in GCA than UCA and was associated with unfavorable prognostic factors. Conversely, UCA cases showed strong p16 expression, supporting their HPV-driven pathogenesis. Molecular profiling revealed *KRAS* and *PIK3CA* mutations in both subtypes, while *BRAF* mutations were identified exclusively in GCA. These findings reveal distinct clinical and molecular profiles for both tumor types and underscore ARID1B and PD-L1 as predictive prognostic and therapeutic biomarkers in GCA, implicating the use of subtype-specific treatment strategies.

## 1. Introduction

Cervical cancer is a significant global health issue, with the fourth-highest incidence and mortality rates among cancers affecting women worldwide [1,2,3,4,5,6,7]. Among its histological subtypes, endocervical adenocarcinoma (ECA) accounts for approximately 20–25% of all cervical cancer cases, and its incidence has been increasing in recent years, particularly in developed countries [8,9,10,11,12,13]. In response to the evolving understanding of cervical cancer biology, the 2020 World Health Organization (WHO) classification introduced a two-tiered system for endocervical adenocarcinomas based on human papillomavirus (HPV) status: HPV-associated and HPV-independent types. Usual-type cervical adenocarcinoma (UCA) is the most prevalent HPV-associated subtype, whereas gastric-type cervical adenocarcinoma (GCA) is the most prominent HPV-independent variant [14,15,16].

Cervical adenocarcinoma (ADC) encompasses a diverse range of morphological subtypes, including usual, villoglandular, mucinous, serous, gastric, clear cell, mesonephric, and endometrioid types, each showing distinct associations with HPV infection. Notably, gastric-type ADC is almost 100% HPV-negative, followed by the endometrioid type (81.1%) [16], whereas UCA demonstrates a strong association with high-risk HPV, particularly types 16 and 18 [17,18], with only 9.6% of cases being HPV-negative [16].

The concept of gastric-type endocervical adenocarcinoma (GAS) was introduced in 2007 [19]. Subsequently, Yin et al. reported that GAS is unrelated to HPV infection and often presents with nonspecific clinical manifestations, making its diagnosis and management particularly challenging [20]. It constitutes a substantial proportion of ECAs in Japan (10–25%) [21,22] and roughly 10% in Western countries [22,23], while its global prevalence is estimated at approximately 3.8% [24], indicating marked regional variation [25,26,27,28,29]. UCA often presents with characteristic cytological features and diffuse p16 expression [30,31,32,33,34]. While GCA poses unique diagnostic and clinical challenges, it tends to show subtle cytologic abnormalities and a deeply infiltrative, endophytic growth pattern, often leading to delayed detection [22,35,36]. Histologically, it is defined by a pale eosinophilic or clear cytoplasm and the expression of gastric-type mucins such as MUC6 and HIK1083 [19,37,38]. Clinically, GCA is associated with more aggressive behavior and a poorer prognosis compared to UCA [39,40,41,42,43].

Molecular profiling has further highlighted the differences between these subtypes. Previously, several targeted sequencing studies of GCA samples have identified a similar broad spectrum of genetic alterations, notably in *TP53*, *CDKN2A/B*, *STK11*, *KRAS*, *ARID1A*, *BRAF*, *ERBB2*, *POLE*, *PIK3CA*, and *GNAS* [44,45,46]. However, UCA more frequently harbored mutations in cell-cycle-related genes, including *TP53*, *PIK3CA*, *KRAS*, *ERBB2/3*, and *ARID1A* [45].

Despite growing recognition of GCA as a clinically and molecularly distinct entity, limited comparative studies have addressed its pathological and molecular features relative to UCA, particularly in the Japanese population, where its incidence is notably higher.

The purpose of this study is to comprehensively compare the clinicopathological, immunohistochemical, and molecular features of GCA and UCA in a Japanese population, with the aim of identifying key prognostic biomarkers and exploring potential therapeutic strategies.

## 2. Results

### 2.1. Clinicopathological Findings of the Patients with UCA or GCA

A total of 110 patients diagnosed with cervical adenocarcinoma were included in this study; 94 patients (85.5%) were diagnosed with UCA and 16 (14.5%) with GCA. UCA cases exhibited HPV-associated morphology, characterized by glandular, papillary, or cribriform architectures lined by pseudostratified columnar cells with prominent apical mitotic figures and apoptotic bodies. In contrast, GCA cases displayed features consistent with HPV-independence, including irregular, lobular glandular structures lined by cells with pale or eosinophilic cytoplasm, minimal nuclear atypia, and infrequent mitotic or apoptotic activity [47,48]. Representative histological features of UCA and GCA in our cohort are illustrated in Figure 1, and the clinicopathological features of the patients are shown in Table 1.

The age at diagnosis ranged from 30 to 75 years, with a median age of 45 years in both the UCA and GCA groups. The mean age across the cohort was comparable between groups, with no statistically significant difference observed (*p* = 0.57). The majority of patients were under 50 years of age, accounting for 66.0% in the UCA group and 56.3% in the GCA group. Detailed staging information revealed notable differences in clinical presentation. A significantly higher proportion of GCA cases were diagnosed at advanced stages (FIGO stage III/IV: 37.5%) compared to UCA cases (13.8%) (*p* = 0.03), reflecting a more aggressive clinical course. Residual tumors were more frequent in GCA patients, with all 16 cases (100%) exhibiting residual disease compared to 87.2% of UCA cases, though this difference did not reach statistical significance.

GCA patients also demonstrated a higher recurrence rate (37.5%) compared to UCA patients (13.8%) (*p* = 0.03). Similarly, overall survival (OS) was significantly worse in the GCA group, where 5 of 16 patients (31.3%) died by the end of the follow-up period, compared to 8 of 94 patients (8.5%) in the UCA group (*p* = 0.02). Lymph node involvement was significantly more frequent in GCA cases, whereas no significant difference was observed in the rate of distant metastases. However, vaginal invasion was observed more often among GCA (37.5%) than UCA patients (13.8%) (*p* = 0.03). These findings underscore the more aggressive and advanced clinical presentation of GCA, further supporting its classification as a distinct subtype with a poorer prognosis.

### 2.2. Survival Analysis

We conducted a Kaplan–Meier analysis to evaluate PFS and OS in the GCA and UCA cohorts (Figure 2).

Patients with GCA exhibited significantly poorer PFS compared to those with UCA (*p* = 0.014). The progression-free survival (PFS) curve for GCA demonstrated an early and continuous decline, particularly within the first two years following treatment, while the UCA group maintained relatively stable PFS throughout the follow-up period. Similarly, OS was also worse in the GCA group compared to the UCA group (*p* = 0.032). The OS curve for GCA patients showed a steady downward trajectory, whereas most UCA patients sustained high survival rates over time. These findings further highlight the aggressive clinical behavior and poorer prognosis associated with GCA in comparison to UCA.

### 2.3. Immunohistochemical Findings

Immunohistochemical analysis was performed to evaluate the expression of p53, p16, and immune escape-related markers (PD-L1, PD-1, and CD8), as well as ARID1A, ARID1B, c-Myc, and PTEN (Table 2).

Aberrant p53 expression, indicative of underlying *TP53* mutations, was significantly more frequent in GCA than in UCA (25% vs. 4.2%, *p* = 0.015). In contrast, p16 overexpression, a surrogate marker of high-risk HPV-associated carcinogenesis, was significantly higher in UCA (87.2%) compared to GCA (43.75%, *p* = 0.0003), reflecting its established HPV-independent nature. Among UCA cases, p16 expression ranged from focal to diffuse ‘block-type’ staining, with most cases exhibiting the latter pattern. For ARID1A, 11 GCA cases (68.75%) showed a wild-type staining pattern, while 5 cases (31.25%) exhibited loss or mutation-type staining. *ARID1A* mutations are generally more common in GCA than in UCA.

PD-L1 positivity was slightly higher in GCA than in UCA, though the difference was not statistically significant (*p* = 0.711). PD-1 expression also showed no significant difference between the groups (*p* = 0.1003). CD8-positive tumor-infiltrating lymphocytes (TILs) tended to be lower in GCA, with 56.25% of cases showing low expression compared to 50% in UCA, though this difference was not statistically significant (*p* = 0.788). Moreover, no significant differences were observed in ARID1B or c-Myc expression between the two groups. Furthermore, PTEN loss tended to be more frequent in GCA than in UCA; however, this difference was not statistically significant.

### 2.4. Molecular Findings

The molecular alterations in the two cohorts were analyzed using two complementary approaches: Sanger sequencing and immunohistochemistry (IHC) (Table 3). Sanger sequencing was performed to detect key oncogenic mutations, especially in *KRAS*, *PIK3CA*, and *BRAF*. *KRAS* mutations were found in 25% of GCA cases and 14.8% of UCA cases. *PIK3CA* mutations were more frequent in UCA (31.6%) than in GCA (7.7%). *BRAF* mutations were identified in 13.3% of GCA cases and 5.3% of UCA cases.

In addition, immunohistochemical (IHC) analysis was conducted to assess the expression patterns of TP53, ARID1A, ARID1B, c-Myc, and PTEN, which may serve as surrogate indicators of underlying genetic alterations or other regulatory changes.

The results of both molecular and immunohistochemical analyses are summarized in Table 3, while previously reported alterations from the literature are provided in Table 4 for reference.

### 2.5. Prognostic Relevance for p16 in UCA

To investigate the prognostic significance of p16 expression in UCA, we assessed its impact on survival outcomes using Kaplan–Meier survival analysis. As noted above, p16 expression was observed in the majority of UCA cases (82/94) and was significantly associated with improved survival outcomes (Figure 3).

We further explored the relationship between p16 expression and clinicopathological features (Table 5).

Positive p16 expression was significantly associated with younger age at diagnosis (<50 years), recurrence, and lower rates of distant metastasis. No significant associations were found with clinical stage, survival status, lymph node involvement, and vaginal invasion. To evaluate whether p16 expression served as an independent prognostic factor, we conducted univariate and multivariate Cox proportional hazards regression analyses for both progression-free survival (PFS) and overall survival (OS) (Appendix A). In the univariate analysis, p16 positivity was associated with favorable PFS and OS. However, in the multivariate model, this association was not statistically significant, suggesting that the prognostic impact of p16 may be influenced by other clinicopathological variables.

Overall, our findings suggest a potential diagnostic and prognostic role for p16 immunoreactivity in UCA, consistent with previous studies that have reported its clinical relevance in HPV-associated cervical adenocarcinomas [51,52].

### 2.6. High ARID1B Expression Predicts Poor Survival and May Be a Potential Prognostic Biomarker in GCA

We performed immunohistochemical and Kaplan–Meier survival analyses to evaluate ARID1B expression in GCA (Figure 4).

The Kaplan–Meier analysis demonstrated that high ARID1B expression was significantly associated with shorter progression-free survival (PFS) and overall survival (OS) in GCA patients. These findings indicate a possible link between elevated ARID1B expression and unfavorable prognosis in this subtype. To further explore its prognostic relevance, we attempted univariate and multivariate Cox regression analyses incorporating ARID1B expression alongside other clinicopathological variables. However, due to a small number of low cases, such analyses were not statistically possible.

### 2.7. PD-L1 Expression in GCA May Serve as a Potential Therapeutic Biomarker for ICI

In our cohort of GCA patients, immunohistochemical analysis revealed PD-L1 positivity in 3 out of 16 cases. Although the number of PD-L1-positive cases was limited, Kaplan–Meier survival analysis demonstrated a clear trend associating PD-L1 expression with shorter progression-free survival (PFS) and overall survival (OS) (Figure 5).

To better understand the clinical relevance of PD-L1 expression, we analyzed its correlation with key clinicopathological parameters. Positive PD-L1 expression was significantly associated with distant metastasis and poorer survival status (*p* < 0.05) (Table 6), which may be a link between PD-L1-mediated immune evasion and more aggressive disease features in GCA. Furthermore, univariate and multivariate Cox regression analyses were performed to assess the prognostic value of PD-L1 expression (Appendix A).

In both univariate and multivariate analyses, PD-L1 indicated an unfavorable prognostic value. Although the number of PD-L1-positive cases in our cohort was small, limiting our ability to confirm its role as an independent prognostic factor, our findings align with and build upon existing evidence.

## 3. Discussion

Cervical adenocarcinoma represents a histologically and molecularly diverse group of malignancies, characterized by distinct etiological pathways and clinical outcomes. Among them, GCA is an uncommon but notably aggressive subtype. Unlike the more prevalent UCA, which is strongly associated with HPV infection and generally responds favorably to standard therapies, GCA arises independently of HPV and exhibits resistance to conventional treatment modalities, resulting in a poorer prognosis [8,40,46].

In this study, we conducted a comparative analysis of GCA and UCA using clinicopathological, immunohistochemical, and molecular data to better characterize their divergent tumor biology and identify potential prognostic markers. Our findings confirmed the aggressive nature of GCA, which was more often associated with advanced FIGO stage, higher rates of recurrence, lymph node involvement, and vaginal invasion. These clinicopathological features translated into significantly worse progression-free survival (PFS) and overall survival (OS) for GCA compared to UCA, consistent with previous reports [47,49,53,54]. Collectively, these data emphasize the biologically and clinically distinct nature of GCA and underline the urgent need for early diagnostic strategies and tailored interventions for this high-risk group.

The survival analysis further illustrated the clinical divergence between the two subtypes, although the number of GCA cases was limited. The patients with GCA had significantly poorer outcomes compared to those with UCA. PFS was significantly lower in the GCA group (*p* = 0.014), with the survival curve showing a sharp decline within the first 30 months post-treatment, followed by a plateau. Alternatively, UCA patients exhibited a more favorable PFS trajectory, with high and stable survival rates maintained throughout the observation period. Similarly, OS was also significantly reduced in GCA patients compared to UCA (*p* = 0.032). The OS curve for GCA dropped earlier and more steeply, while UCA patients showed sustained long-term survival with minimal mortality over time. These findings also highlight the aggressive nature and unfavorable prognosis of GCA relative to UCA, in line with previous studies reporting similar trends in clinical outcomes for GCA [22,28,50].

To explore the underlying molecular mechanisms, we performed targeted Sanger sequencing for selected oncogenes. Our data revealed that *KRAS* mutations were more prevalent in GCA, consistent with prior reports suggesting its oncogenic involvement in HPV-independent carcinogenesis [45,46]. *BRAF* mutations, although less frequent, were also identified in GCA, supporting the occasional activation of the MAPK pathway in this subtype [45]. Conversely, *PIK3CA* mutations were more prevalent in UCA, in line with TCGA data showing frequent activation of the PI3K/AKT pathway in HPV-associated tumors [45,46,55]. Immunohistochemical analysis provided further insight into the molecular underpinnings of GCA and UCA. GCA was frequently negative or only patchily positive for p16, consistent with its lack of HPV-driven oncogenesis [56]. In contrast, UCA exhibited strong and diffuse p16 staining, a hallmark of high-risk HPV-mediated transformation via E7-mediated RB1 inactivation [57]. The Kaplan–Meier survival analysis confirmed that p16-positive UCA cases had better OS, reinforcing p16’s value as both a diagnostic and prognostic marker in HPV-associated adenocarcinomas [51,52].

The staining pattern of p53 also differed notably between the subtypes. GCA showed strong, diffuse nuclear p53 expression indicative of *TP53* mutations [48], while UCA typically exhibited wild-type expression, characterized by weak or absent nuclear staining. These reciprocal patterns reflect distinct molecular pathogenesis. In HPV-negative GCA, *TP53* mutations may lead to p53 accumulation, causing intact RB1 expression, explaining the low p16 expression. On the other hand, HPV-positive UCA shows p16 overexpression, probably due to HPV E7-induced RB inhibition, while wild-type *p53* may be inactivated by HPV E6, not by gene mutations causing overexpression [4,19,58,59].

Notably, we identified elevated ARID1B expression in GCA, which was significantly associated with poor survival outcomes. This trend suggests a contributory role of ARID1B in tumor aggressiveness, possibly via chromatin remodeling or transcriptional regulation. Wang et al. reported that in bladder urothelial carcinoma, immunohistochemical analyses have shown that elevated ARID1B levels correlate with significantly shorter OS and PFS [60]. Multivariate analysis further confirmed ARID1B as an independent predictor of poor OS in these patients. Similarly, in breast cancer, ARID1B overexpression was associated with worse outcomes via disruption in the SWI/SNF complex [61,62]. Another study of invasive ductal breast carcinoma showed that high ARID1B expression is linked to significantly lower 5-year disease-free survival compared to tumors with low expression [63]. ARID1B functions as a mutually exclusive paralog of ARID1A within the SWI/SNF (BAF) chromatin-remodeling complex. Loss of ARID1A often leads to compensatory upregulation of ARID1B to maintain complex function. This phenomenon is observed in both clinical samples and in vitro studies [64]. Helming et al. reported that nearly all ARID1A-mutant cancers retain at least one intact ARID1B allele, indicating selective pressure to preserve ARID1B for cell viability. Thus, ARID1A-deficient tumors become increasingly dependent on ARID1B for chromatin remodeling [65,66]. This compensatory ARID1B upregulation is commonly seen in poorly differentiated, aggressive cancers, such as colorectal tumors, where ARID1A expression is low and ARID1B is elevated [65]. Mechanistically, this may occur through promoter hypomethylation or transcriptional activation by oncogenic signals. Notably, ARID1A and ARID1B can exert opposing effects on cell-cycle gene regulation, which may contribute to the aggressive phenotype associated with high ARID1B expression [60].

To our knowledge, this is the first report to evaluate ARID1B expression in GCA. Given the limited number of GCA cases, our findings were insufficient to determine whether ARID1B serves as an independent prognostic factor.

We also identified PD-L1 positive expression correlated with adverse features, including lymph node and distant metastasis, and significantly poorer survival outcomes, supporting the hypothesis that PD-L1 contributes to immune escape mechanisms in GCA [67,68,69]. Although we did not directly assess immunotherapy efficacy, previous studies demonstrated high PD-L1 expression as a predictive biomarker for immune checkpoint inhibitor-based therapies in GCA [67,68,69]. Additionally, while immunotherapy significantly prolonged OS in recurrent or metastatic GCA, its effect on PFS was limited, likely influenced by subsequent treatments [67,69,70]. In parallel, in studies on gastric cancers, patients whose tumors were both PD-L1 positive and rich in CD8^+^ TILs had improved survival compared to those lacking one or both factors [71]. This suggests a synergistic interaction between immune markers: the presence of PD-L1 indicates an active T-cell response (albeit inhibited), and blocking the PD-1/PD-L1 axis can unleash these T-cells’ activity. Indeed, co-assessment of PD-L1 status and CD8^+^ TIL density has been proposed as a more informative predictor of ICI therapy efficacy than PD-L1 alone [72]. In essence, PD-L1 does not act in isolation but as part of a dynamic immune contexture, and its prognostic and therapeutic significance in GCA is best understood in light of tumor-infiltrating immune cells (CD8^+^ effector T-cells and even FOXP3^+^ regulatory T-cells) that collectively shape the tumor’s response to immune checkpoint blockade [71,72]. By recognizing these synergistic effects between immune markers, future strategies could combine PD-1/PD-L1 inhibitors with treatments that enhance T-cell infiltration or relieve other immunosuppressive mechanisms, thereby potentiating the overall anti-tumor immune response. Although the number of PD-L1-positive cases in our Japanese cohort is small (n = 3), our findings are consistent with prior studies and support the hypothesis that PD-L1 may contribute to immune escape in GCA.

This study addresses certain limitations. The primary limitation of this study was the relatively small number of gastric-type adenocarcinoma (GCA) cases (n = 16), which reflects the inherent rarity of this tumor subtype. While we identified clinically and statistically meaningful associations, particularly involving ARID1B and PD-L1 expression, the limited sample size may have constrained the statistical power for certain subgroup analyses. Another limitation was that we identified key oncogenic mutations through Sanger sequencing instead of next-generation sequencing. Despite these limitations, our findings are consistent with prior reports and offer novel insight into the molecular and immunological landscape of GCA, particularly within a Japanese population. Future validation using larger, multi-institutional cohorts will be essential to confirm these observations and further clarify the prognostic and therapeutic relevance of the biomarkers investigated.

## 4. Materials and Methods

### 4.1. Specimen Acquisition

This collaborative study included a total of 110 cervical cancer tissue samples collected from two institutions: the Department of Obstetrics and Gynecology at Shimane University School of Medicine (Shimane, Japan) between 2014 and 2017 and Seirei Hamamatsu Hospital (Shizuoka, Japan) between 2003 and 2022. Of these, 94 samples were collected from patients diagnosed with UCA, and 16 were from patients with GCA. All diagnoses were reviewed and confirmed by two gynecologic pathologists at our institution. Relevant clinicopathological data were extracted from medical records and pathological reports.

### 4.2. Ethical Approval and Consent

This study was approved by the Institutional Review Board of the Shimane University School of Medicine (IRB Nos. 20070305-1 and 20070305-2), and after receiving a detailed explanation of the procedure and the study, all patients provided written informed consent.

### 4.3. Immunohistochemistry

Formalin-fixed, paraffin-embedded (FFPE) tissue blocks were sectioned at a thickness of 5 μM [73,74]. FFPE slides were then deparaffinized and rehydrated with xylene and alcohol solutions. Antigen retrieval was performed using either Tris-EDTA buffer or sodium citrate buffer with a pH of 9 (Dako, Glostrup, Denmark), depending on the antibody, followed by cooling at room temperature. Endogenous peroxidase activity was blocked with 0.3% H_2_O_2_. The slides were then incubated at 4 °C for overnight with primary antibodies. The primary antibodies used were p53, p16, PD-L1, PD-1, CD8, ARID1A, ARID1B, c-Myc, and PTEN. Details regarding antibody clones, dilutions, and sources are provided in Appendix A. Following primary antibody incubation, the slides were washed and treated with the appropriate secondary antibodies in a humidified chamber at room temperature. Chromogenic detection was performed using 3,3′-diaminobenzidine (DAB), followed by counterstaining with Mayer’s hematoxylin; after dehydration and clearing, the slides were sealed with coverslips. Appropriate positive controls were included in each staining batch to validate the procedure. All stained sections were examined under a light microscope. Immunoreactivity was independently evaluated by an experienced pathologist (MN) blinded to the clinicopathological data.

Immunohistochemistry (IHC) scoring was performed using a semi-quantitative H-score approach, evaluating both the staining intensity and the percentage of tumor cells positive in the nuclei and cytoplasm. Staining intensity was graded on a 0 to 3+ scale: 0 indicated no detectable staining, 1+ weak staining (involving up to ~30% of cells), 2+ moderate staining (~30–50% of cells), and 3+ strong staining (≥60% of cells) [75,76]. For p16, combined nuclear and cytoplasmic immunoreactivity was classified as p16 positive (2+ and 3+, reflecting diffuse strong p16 staining), which was consistent with prior reports defining positivity as strong nuclear/cytoplasmic p16 staining in >70% of tumor cells, and p16 negative (0 and 1+ staining). The density of CD8 tumor-infiltrating lymphocytes (TILs) was categorized as negative (0+ or 1+) or positive (2+ or 3+), with ≥30% stromal infiltration regarded as CD8 positivity based on previous studies [57]. The PD-L1 and PD-1 IHC results were interpreted as positive if ≥5% of cells were stained (applying to tumor cells for PD-L1 and to TILs for PD-1) [77]. The p53 IHC pattern was thought to be wild type when staining was heterogeneous (score 1+ or 2+, roughly 10–50% of tumor cells positive) and as mutant type when p53 was either diffusely overexpressed (score 3+, ≥60% of cells) or completely absent (0%). ARID1A protein expression was considered wild type if any tumor cell staining was present (≥10% of cells positive), whereas complete loss of ARID1A (0% staining) indicated the mutation type. PTEN was assessed in the same manner: cases with any detectable PTEN expression were categorized as retained and those with no staining as PTEN loss. Finally, ARID1B and c-Myc expression levels were classified as high when scored 3+ (staining in ≥60% of tumor cells) and low when scored 0–2+ (staining in <60% of cells).

### 4.4. Genetic Analyses by Sanger Sequencing

Genomic DNA was extracted from macrodissected FFPE tissue samples targeting macroscopically apparent tumor regions. Mutational analysis was conducted for *KRAS*, *BRAF*, and *PIK3CA* in all samples. Targeted regions included exons 2, 9, and 15 of *KRAS*, *PIK3CA*, *BRAF*, respectively, which are known hotspots for oncogenic mutations [78,79,80,81]. Polymerase chain reaction (PCR) amplification was carried out using exon-specific primers. The thermal cycling conditions included an initial denaturation at 95 °C for 30 s, followed by 40 cycles of annealing at 55 °C and extension at 72 °C for 15 s. All PCR products were subjected to direct sequencing using the Sanger method. Sequencing was conducted at Beckman Coulter (Danvers, MA, USA), and data were analyzed using the Mutation Surveyor DNA Variant Analysis Software company (Tokyo, Japan). Primer sequences of the current study were synthesized by GeneLink, Inc. (Hawthorne, NY, USA) and are listed in Appendix A, as previously described [82]. The pathogenicity of detected variants was confirmed using the Catalogue of Somatic Mutations in Cancer (COSMIC) database [83]. All procedures adhered to standardized protocols to ensure their accuracy and reproducibility.

### 4.5. Statistical Analysis

Statistical analyses were performed to compare the clinicopathological characteristics of GCA and UCA. Fisher’s exact test was used to evaluate categorical associations, while Pearson’s chi-squared test was applied for correlation analysis. Progression-free survival (PFS) and overall survival (OS) were estimated using the Kaplan–Meier method, and survival differences between groups were evaluated with the log-rank test. Univariate and multivariate analyses were conducted using Cox proportional hazards regression models, with binomial logistic regression applied to explore the relationships between clinicopathological variables and survival outcomes. All statistical analyses were conducted using IBM SPSS Statistics for Windows, version 27.0.1 (IBM Corp., Armonk, NY, USA), and a two-tailed *p*-value <0.05 was considered statistically significant.

## 5. Conclusions

This comparative study underscores distinct differences between GCA and UCA in clinical presentation, molecular alterations, and immune profiles. GCA demonstrated more aggressive features and poorer survival than UCA. Immunohistochemically, GCA showed p16 negativity and aberrant p53 accumulation, supporting an HPV-independent pathway, whereas UCA exhibited strong p16 expression and wild-type p53, consistent with an HPV-associated pathway. Furthermore, *KRAS* and *BRAF* mutations were more frequent in GCA, while *PIK3CA* alterations predominated in UCA. Remarkably, ARID1B and PD-L1 expression in GCA were linked to adverse features and poorer survival, indicating possible prognostic relevance. Overall, these findings provide clinical implication for subtype-specific molecular and immunotherapeutic strategies in cervical adenocarcinoma.

## Figures and Tables

**Figure 1 ijms-26-07469-f001:**
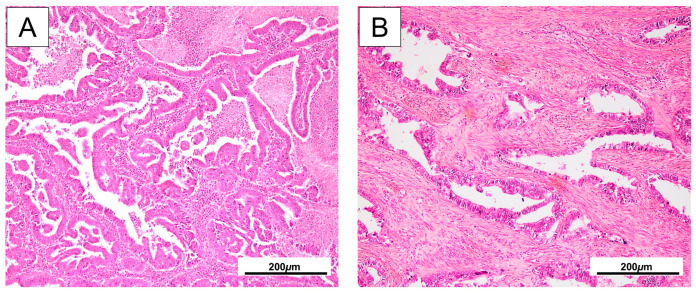
Representative H&E-stained sections of UCA and GCA. (**A**) UCA displaying well-formed glandular structures lined by atypical columnar epithelial cells with pseudostratified, hyperchromatic nuclei and frequent mitoses. (**B**) GCA is characterized by irregular, dilated glands with pale to eosinophilic cytoplasm, distinct cell borders, and variably atypical nuclei, often with clear or foamy appearance.

**Figure 2 ijms-26-07469-f002:**
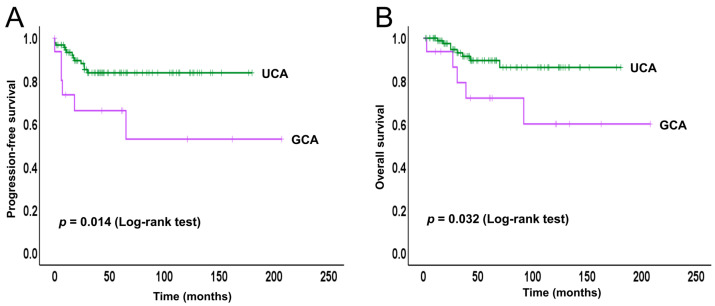
Kaplan–Meier curves for the patients with GCA and UCA. (**A**) Progression-free survival (PFS) showing significantly poorer outcomes in the GCA group compared to UCA (log-rank test, *p* = 0.014). (**B**) Overall survival (OS) indicating a worse prognosis in GCA than UCA (log-rank test, *p* = 0.032), with a steady decline observed in the GCA cohort.

**Figure 3 ijms-26-07469-f003:**
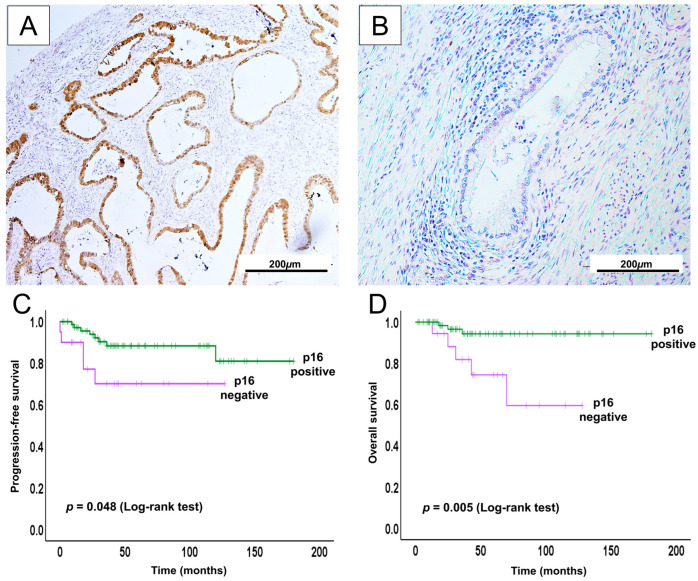
Prognostic significance of p16 expression in UCA. (**A**,**B**) Representative immunohistochemical staining of p16 showing positive (**A**) and negative (**B**) expressions. Kaplan–Meier curves indicating that p16-positive cases had significantly better progression-free survival (**C**) (log-rank test, *p* = 0.048) and overall survival (**D**) (log-rank test, *p* = 0.005) compared to p16-negative cases.

**Figure 4 ijms-26-07469-f004:**
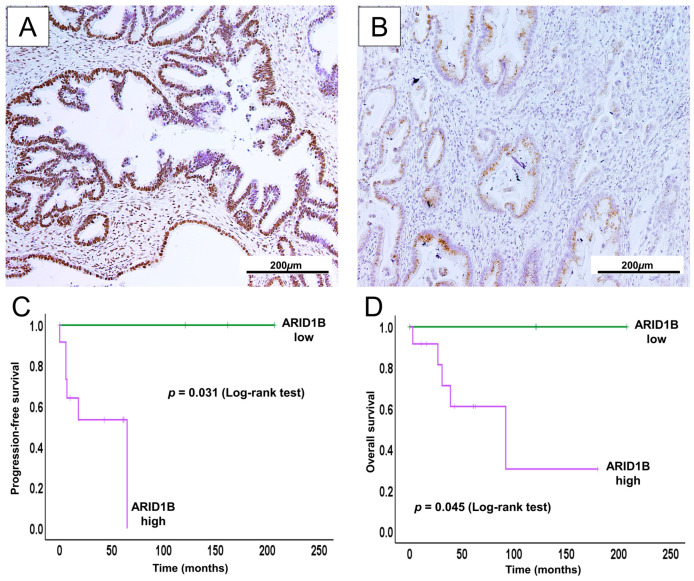
Immunohistochemical expression of ARID1B and its association with clinical outcomes in GCA. Representative high (**A**) and low (**B**) ARID1B nuclear staining patterns. (**C**) High ARID1B expression was associated with significantly worse progression-free survival (log-rank test, *p* = 0.031). (**D**) Similarly, overall survival was poor in patients with high ARID1B expression (log-rank test, *p* = 0.045).

**Figure 5 ijms-26-07469-f005:**
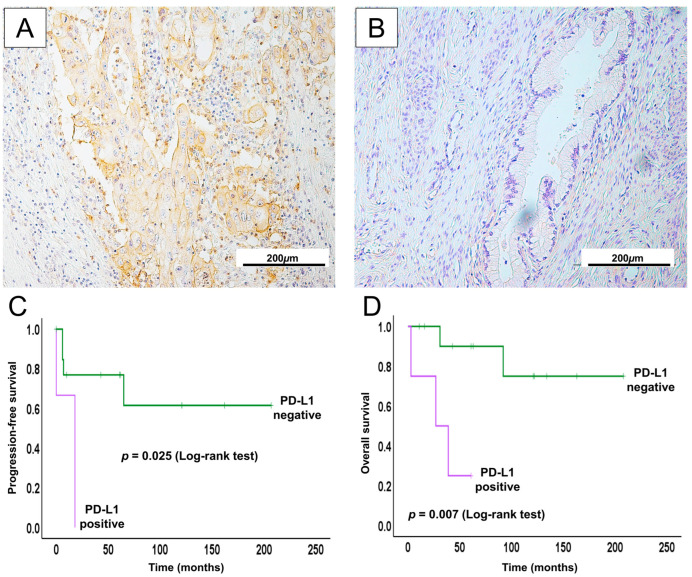
PD-L1 expression in GCA and its potential clinical significance. Representative positive (**A**) and negative (**B**) PD-L1 expression patterns. Kaplan–Meier curves demonstrate that positive or high PD-L1 expression correlates with significantly poorer (**C**) progression-free survival (log-rank test, *p* = 0.025) and (**D**) overall survival (log-rank test, *p* = 0.007).

**Table 1 ijms-26-07469-t001:** Clinicopathological features of patients with usual- and gastric-type cervical adenocarcinoma.

Characteristics	UCA	GCA	*p*-Value
**No. of patients**	94	16	
**Age**			0.57
<50 years	62 (66%)	9 (56.25%)	
≥50 years	32 (34%)	7 (43.75%)	
**Clinical FIGO stage**			0.03 *
AIS	12 (12.8%)	0 (0%)	
I, II	69 (73.4%)	10 (62.5%)	
III, IV	13 (13.8%)	6 (37.5%)	
**Residual tumor**			0.2
No	12 (12.8%)	0 (0%)	
Yes	82 (87.2%)	16 (100%)	
**Recurrence**			0.03 *
Yes	13 (13.8%)	6 (37.5%)	
No	81 (86.2%)	10 (62.5%)	
**Survival status**			0.02 *
Death	8 (8.5%)	5 (31.25%)	
Alive	86 (91.5%)	11 (68.75%)	
**Lymph node involvement**			0.47
Yes	15 (16%)	4 (25%)	
No	79 (84%)	12 (75%)	
**Distant metastasis**			0.37
Yes	2 (2.1%)	1 (6.25%)	
No	92 (97.9%)	15 (93.75%)	
**Vaginal invasion**			0.03 *
Yes	13 (13.8%)	6 (37.5%)	
No	81 (86.2%)	10 (62.5%)	

GCA: gastric-type cervical adenocarcinoma; UCA: usual-type cervical adenocarcinoma; AIS: adenocarcinoma in situ; *: indicates statistically significant (*p* < 0.05).

**Table 2 ijms-26-07469-t002:** Percentage of protein expressions in GCA and UCA cases.

Protein Names	UCA	GCA	*p*-Value
**No. of patients**	94	16	
**p53**			0.015 *
Wild type	90 (95.8%)	12 (75%)	
Mutation type	4 (4.2%)	4 (25%)	
**p16**			0.0003 *
Negative	12 (12.8%)	9 (56.25%)	
Positive	82 (87.2%)	7 (43.75%)	
**PD-L1**			0.711
Negative	80 (85.1%)	13 (81.25%)	
Positive	14 (14.9%)	3 (18.75%)	
**PD-1**			0.1003
Negative	82 (87.2%)	14 (87.5%)	
Positive	12 (12.8%)	2 (12.5%)	
**CD8**			0.788
Negative	47 (50%)	9 (56.25%)	
Positive	47 (50%)	7 (43.75%)	
**ARID1A**			0.14
Wild type	80 (85.1%)	11 (68.75%)	
Mutation type	14 (14.9%)	5 (31.25%)	
**ARID1B**			0.27
Low	1 (1.1%)	1 (6.25%)	
High	93 (98.9%)	15 (93.75%)	
**c-Myc**			1
Low	35 (37.2%)	6 (37.5%)	
High	59 (62.8%)	10 (62.5%)	
**PTEN**			0.085
Loss	26 (27.7%)	8 (50%)	
Retain	68 (72.3%)	8 (50%)	

GCA: gastric-type cervical adenocarcinoma; UCA: usual-type cervical adenocarcinoma; *: indicates statistically significant (*p* < 0.05).

**Table 3 ijms-26-07469-t003:** Summary of molecular alterations and immunohistochemical profiles of mutation markers in gastric and usual-type cervical adenocarcinoma.

Groups	*KRAS*	*PIK3CA*	*BRAF*	TP53 (IHC)	ARID1A (IHC)	ARID1B (IHC)	c-Myc (IHC)	PTEN (IHC)
GCA	25%	7.7%	13.3%	25%	68.75%	93.75%	37.5%	50%
UCA	14.8%	31.6%	5.3%	4.2%	33%	98.9%	32%	27.7%

IHC: immunohistochemistry; GCA: gastric-type cervical adenocarcinoma; UCA: usual-type cervical adenocarcinoma.

**Table 4 ijms-26-07469-t004:** Previously reported molecular alterations in the literature.

Author Name	*KRAS*	*PIK3CA*	*BRAF*	*TP53*	*ARID1A*	c-Myc Amp.	*PTEN*	Reference
	GCA UCA	GCA UCA	GCA UCA	GCA UCA	GCA UCA	GCA UCA	GCA UCA	
Park E., et.al.	4.8% –	– –	– –	52.4% –	– –	– –	– –	[2]
Nasu H., et. al.	12.5% –	25% –	– –	– –	– –	– –	– –	[25]
Selenica P., et. al.	18% 11%	7% 25%	4% –	41% 5%	6% 10%	– –	– –	[45]
Hodgson A., et. al.	36% 13%	18% 16%	– –	46% 11%	– –	– 4%	– –	[46]
Lu S., et. al.	13.3% –	– –	– –	53.3% –	20% –	– –	20% –	[49]
Ehmann S., et.al.	30% –	15% –	– –	74% –	– –	– –	– –	[50]

Amp: amplification; GCA: gastric-type cervical adenocarcinoma; UCA: usual-type cervical adenocarcinoma.

**Table 5 ijms-26-07469-t005:** Relationship between p16 and clinicopathological parameters of patients with usual-type cervical adenocarcinoma.

Clinicopathological Parameters	p16 Positive (n = 74)	p16 Negative (n = 20)	*p*-Value
**Age**			0.011 *
<50 years	57 (77%)	9 (45%)	
≥50 years	17 (23%)	11 (55%)	
**Clinical FIGO stage**			0.37
I, II	59 (79.7%)	14 (70.0%)	
III, IV	15 (20.3%)	6 (30.0%)	
**Recurrence**			0.0014 *
Yes	8 (10.8%)	9 (45.0%)	
No	66 (89.2%)	11 (55.0%)	
**Survival status**			0.06
Death	2 (2.7%)	3 (15.0%)	
Alive	72 (97.3%)	17 (85.0%)	
**Lymph node involvement**			0.45
Yes	11 (14.9%)	1 (5.0%)	
No	63 (85.1%)	19 (95.0%)	
**Distant metastasis**			0.0015 *
Yes	68 (91.9%)	12 (60.0%)	
No	6 (8.1%)	8 (40.0%)	
**Vaginal invasion**			0.29
Yes	64 (86.5%)	15 (75.0%)	
No	10 (13.5%)	5 (25.0%)	

*: Indicates statistically significant (*p* < 0.05).

**Table 6 ijms-26-07469-t006:** Relationship between PD-L1 and clinicopathological parameters of patients with gastric-type cervical adenocarcinoma.

Clinicopathological Parameters	PD-L1 Negative (n = 13)	PD-L1 Positive (n = 3)	*p*-Value
**Age**			1
<50 years	7 (53.8%)	2 (66.7%)	
≥50 years	6 (46.2%)	1 (33.3%)	
**Clinical FIGO stage**			1
I, II	8 (61.5%)	2 (66.7%)	
III, IV	5 (38.5%)	1 (33.3%)	
**Recurrence**			0.51
Yes	4 (30.8%)	2 (66.7%)	
No	9 (69.2%)	1 (33.3%)	
**Survival status**			0.01 *
Death	2 (15.4%)	3 (100%)	
Alive	11 (84.6%)	0 (0%)	
**Lymph node involvement**			0.52
Yes	4 (30.7%)	0 (0%)	
No	9 (69.2%)	3 (100%)	
**Distant metastasis**			0.02 *
Yes	0 (0%)	2 (66.7%)	
No	13 (100%)	1 (33.3%)	
**Vaginal invasion**			1
Yes	5 (38.5%)	1 (33.3%)	
No	8 (61.5%)	2 (66.7%)	

*: Indicates statistically significant (p < 0.05).

## Data Availability

The data presented in this study are available on request from the corresponding authors.

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
