# Peer review of "A Comparative Analysis of Usual- and Gastric-Type Cervical Adenocarcinoma in a Japanese Population Reveals Distinct Clinicopathological and Molecular Features with Prognostic and Therapeutic Insights"

_ijms, 2025, doi:10.3390/ijms26157469_

Round 1

Reviewer 1 Report

Comments and Suggestions for Authors

This manuscript presents a retrospective comparative analysis of gastric-type cervical adenocarcinoma (GCA) and usual-type cervical adenocarcinoma (UCA) in a Japanese population. The authors provide a detailed clinicopathological, immunohistochemical, and molecular profile of 110 cases (16 GCA, 94 UCA), identifying ARID1B and PD-L1 as potential prognostic and therapeutic biomarkers for GCA. The study is timely and contributes to the growing literature distinguishing HPV-associated and HPV-independent cervical adenocarcinomas.

The GCA cohort is limited (n=16), significantly affecting the robustness of statistical comparisons. While the authors acknowledge this, the conclusions drawn—especially regarding biomarker significance—need tempering. Multivariate analysis for ARID1B was not feasible due to limited sample size. Conclusions regarding its independent prognostic value should be framed more cautiously.

PD-L1 expression was observed in only 3 of 16 GCA cases, yet a survival difference was claimed. The extremely small “positive” cohort (n=3) is insufficient to support the conclusion that PD-L1 is an independent prognostic factor. This should be discussed as hypothesis-generating rather than conclusive. Several results approaching significance (e.g., PTEN loss in GCA vs. UCA) are interpreted suggestively despite not reaching p < 0.05. Authors should avoid over-interpretation and clearly distinguish statistical significance from trends.

This study offers valuable clinicopathologic and molecular insights into cervical adenocarcinoma subtypes in a Japanese population, with particular emphasis on the aggressive nature of GCA. However, the small GCA sample size, limited mutational analysis approach, and overinterpretation of marginal findings should be addressed prior to publication.

Ensure all figures, especially histology and Kaplan–Meier plots, are of high resolution with legible legends. Also, verify that supplementary tables (S3–S6) are properly cited, labeled, and accessible. Clear visuals and data transparency enhance the perceived quality of the work and reduce reviewer skepticism, even when the data are limited.

Comments on the Quality of English Language

To meet publication standards in IJMS, the manuscript would benefit from language polishing by a native English-speaking editor or a professional scientific editing service.

Author Response

We sincerely thank the reviewer for the thoughtful summary and constructive comments, which have greatly assisted us in improving the clarity and scientific rigor of our manuscript. We are grateful for your recognition of the significance and relevance of our study and for your positive assessment of our work and contribution to the understanding of cervical adenocarcinoma subtypes. Below, we provide detailed responses to each point and outline the corresponding revisions made to the manuscript.

Reviewer comment 1:

The GCA cohort is limited (n=16), significantly affecting the robustness of statistical comparisons. While the authors acknowledge this, the conclusions drawn—especially regarding biomarker significance—need tempering. Multivariate analysis for ARID1B was not feasible due to limited sample size. Conclusions regarding its independent prognostic value should be framed more cautiously.

Response 1:

Thank you for raising this valuable concern. We fully agree with the reviewer that the limited number of GCA cases may reduce the statistical power of certain analyses. Accordingly, we have revised the relevant result sections and other portions to clearly state that findings regarding ARID1B should be interpreted as hypothesis-generating. We also clarified that multivariate analysis was not statistically feasible due to the small number of ARID1B-low cases.

Reviewer comment 2:

PD-L1 expression was observed in only 3 of 16 GCA cases, yet a survival difference was claimed. The extremely small “positive” cohort (n=3) is insufficient to support the conclusion that PD-L1 is an independent prognostic factor. This should be discussed as hypothesis-generating rather than conclusive.

Response 2:

We appreciate this important observation. We have revised the text in all sections where it was discussed to explicitly acknowledge the limited number of PD-L1-positive cases and to reframe our findings as exploratory.

Reviewer comment 3:

Several results approaching significance (e.g., PTEN loss in GCA vs. UCA) are interpreted suggestively despite not reaching p < 0.05. Authors should avoid overinterpretation and clearly distinguish statistical significance from trends.

Response 3:

Thank you for this valuable comment. We have revised relevant portions of the results to ensure a clear distinction between statistically significant findings and non-significant trends.

Reviewer comment 4:

This study offers valuable clinicopathologic and molecular insights into cervical adenocarcinoma subtypes in a Japanese population, with particular emphasis on the aggressive nature of GCA. However, the small GCA sample size, limited mutational analysis approach, and overinterpretation of marginal findings should be addressed prior to publication.

Response 4:

We sincerely thank the reviewer for this thoughtful comment. We acknowledge the limitations stemming from the small GCA sample size and the breadth of the mutational analysis, which were previously addressed in the discussion. To improve clarity, we have now further refined and expanded the dedicated limitations section, ensuring a more balanced interpretation of our findings. Additionally, we have carefully re-examined the data and revised the language throughout the manuscript to prevent any potential overinterpretation of marginal results. We believe these updates have further enhanced the scientific rigor and overall quality of the manuscript.

Reviewer comment 5:

Ensure all figures, especially histology and Kaplan–Meier plots, are of high resolution with legible legends. Also, verify that supplementary tables (S3–S6) are properly cited, labeled, and accessible. Clear visuals and data transparency enhance the perceived quality of the work and reduce reviewer skepticism, even when the data are limited.

Response 5:

We are grateful to the reviewer for highlighting this important point. We have carefully reviewed and updated all figures to ensure high resolution and clarity. Figure legends have also been revised for improved readability and completeness. Additionally, we confirmed that all supplementary tables (S3–S6) are properly cited within the main text and are labeled and formatted according to journal guidelines. We thank the reviewer for emphasizing the importance of visual quality and data transparency.

*** Once again we would like to express our gratitude to the reviewer for the constructive feedback, which has led to substantial improvements in the manuscript.

Reviewer 2 Report

Comments and Suggestions for Authors

1. The study focuses on a comparative analysis of gastric-type cervical adenocarcinoma (GCA) and usual-type cervical adenocarcinoma (UCA), systematically investigating the differences in clinicopathological features, immunohistochemical markers, and molecular mutations between the two subtypes in a Japanese population, providing valuable insights for subtype-specific treatment strategies. 2. Through a retrospective analysis of 110 cases, combined with Kaplan-Meier and Cox regression analyses, the study clearly demonstrates that GCA is more aggressive (e.g., higher lymph node metastasis rate, advanced FIGO stage, recurrence rate, and poorer survival rate) and identifies ARID1B and PD-L1 as potential prognostic and therapeutic biomarkers. The research design is rigorous, and the conclusions have clinical reference significance. 3. Molecular analysis reveals differences in KRAS and PIK3CA mutations between the two subtypes, as well as the specificity of BRAF mutations in GCA, providing molecular evidence for understanding their carcinogenic mechanisms. Suggestions for Improvement: 1. The study mentions that GCA is an HPV-independent tumor, however, the Japanese population does not represent other regional populations. It is recommended to refer to and cite relevant literature.(doi:10.1136/ gocm-2024-000160 ),which systematically reviews the prevalence, pathological, and molecular characteristics of HPV-negative cervical cancers, specifically pointing out that gastric-type cervical adenocarcinoma is a major subtype of HPV-negative cervical cancer with variations among different populations, to strengthen the consensus on the classification of HPV-negative cervical cancer subtypes, and to supplement and compare the differences in HPV-negative rates of GCA among different populations, thereby enhancing the horizontal relevance of the research. 2. The sample size of GCA is only 16 cases. Although this is consistent with its rarity, it may limit the power of some statistical analyses (e.g., insufficient cases with low ARID1B expression leading to the inability to perform multivariate analysis). It is recommended to clearly state this limitation in the discussion and further validate the findings with larger sample sizes from multi-center studies. 3. Regarding the therapeutic significance of PD-L1 in GCA, only its association with immune checkpoint inhibitor (ICI) therapy is mentioned, but the correlation with the tumor microenvironment (e.g., CD8+ T cell infiltration) is not explored in depth. Supplementary analysis of the synergistic effects between immune markers can be added to improve the discussion on the potential mechanisms of ICI therapy.

Author Response

We would like to sincerely thank the reviewer for the positive evaluation of our manuscript and for highlighting the scientific and clinical relevance of our findings. We appreciate the thoughtful suggestions which have helped improve the quality and depth of our study. Below are our point-by-point responses to each comment, with corresponding revisions noted.

Reviewer comment 1:

The study mentions that GCA is an HPV-independent tumor, however, the Japanese population does not represent other regional populations. It is recommended to refer to and cite relevant literature.(doi:10.1136/gocm-2024-000160),which systematically reviews the prevalence, pathological, and molecular characteristics of HPV-negative cervical cancers, specifically pointing out that gastric-type cervical adenocarcinoma is a major subtype of HPV-negative cervical cancer with variations among different populations, to strengthen the consensus on the classification of HPV-negative cervical cancer subtypes, and to supplement and compare the differences in HPV-negative rates of GCA among different populations, thereby enhancing the horizontal relevance of the research.

Response 1:

Thank you for this insightful comment. We have revised the introduction to include reference to the suggested article.

Reviewer comment 2:

The sample size of GCA is only 16 cases. Although this is consistent with its rarity, it may limit the power of some statistical analyses (e.g., insufficient cases with low ARID1B expression leading to the inability to perform multivariate analysis). It is recommended to clearly state this limitation in the discussion and further validate the findings with larger sample sizes from multi-center studies.

Response 2:

We fully acknowledge this limitation and had previously addressed it in the discussion; however, we have further revised and expanded the dedicated limitations paragraph to ensure greater clarity and understanding.

Reviewer comment 3:

“Regarding the therapeutic significance of PD-L1 in GCA, only its association with immune checkpoint inhibitor (ICI) therapy is mentioned, but the correlation with the tumor microenvironment (e.g., CD8+ T cell infiltration) is not explored in depth. Supplementary analysis of the synergistic effects between immune markers can be added to improve the discussion on the potential mechanisms of ICI therapy.”

Response 3:

We appreciate this important suggestion. Although our data did not allow in-depth immune profiling, we have revised the PD-L1 section of the discussion to incorporate relevant published findings on the tumor microenvironment, and relevant studies were cited to support this expanded discussion.

*** Once again, we sincerely thank the reviewer for their valuable feedback. We believe these revisions have improved the depth and clarity of the manuscript and better contextualized our findings within the broader field of HPV-independent cervical adenocarcinoma research.